# Left Prefrontal tDCS during Learning Does Not Enhance Subsequent Verbal Episodic Memory in Young Adults: Results from Two Double-Blind and Sham-Controlled Experiments

**DOI:** 10.3390/brainsci13020241

**Published:** 2023-01-31

**Authors:** Gergely Bartl, Paul Allen, Marco Sandrini

**Affiliations:** 1Division of Psychiatry, Faculty of Brain Sciences, University College London, London W1T 7BN, UK; 2School of Psychology, University of Roehampton, London SW15 4JD, UK; 3Department of Psychosis Studies, Institute of Psychiatry, Psychology and Neuroscience, King’s College London, London SE5 8AF, UK

**Keywords:** transcranial direct current stimulation, memory enhancement, verbal episodic memory, encoding, learning and memory

## Abstract

Recent studies suggest that transcranial direct current stimulation (tDCS) applied over the prefrontal cortex (PFaC) may enhance episodic memory ability. As such, there is ongoing interest in the therapeutic potential of this technique in age-related memory decline. At the same time, the findings are not yet conclusive regarding the magnitude of this effect, and assumptions regarding underlying brain mechanisms of stimulation-induced changes in behaviour are yet to be tested in detail. Here, we evaluated the effect of tDCS over left PFC on verbal episodic memory in young adults. Two separate randomized, double-blind, sham-controlled experiments were carried out using (1) incidental learning followed by a recognition test and (2) intentional learning followed by a free recall. In both studies, participants performed a learning task with active or sham tDCS during the encoding period, followed by retrieval tasks on the same day and the next day. The results suggest that, contrary to expectations, active tDCS did not enhance memory performance relative to sham tDCS. Possible reasons behind the lack of enhancement effects are discussed, including the possibility that memory enhancement effects of tDCS may be smaller than first thought. Scientific practices that could improve estimation accuracy in the field are also discussed.

## 1. Introduction

A number of studies have suggested that transcranial direct current stimulation (tDCS) [1,2] has the potential to modulate neural activity and associated cognitive function. Some previous studies have reported promising enhancement in subjects’ retrieval of encoded material when tDCS was applied in either the learning and/or the recognition phase [3,4,5,6,7]. In this paper, our interest is in research relating to the left lateral prefrontal cortex (PFC) as this has been shown to play a causal role in the encoding of episodic memories [8,9,10], and is a frequently studied stimulation target area when aiming to enhance verbal memory. We considered studies measuring subsequent recognition and recall to evaluate the effectiveness of (a) a single session of tDCS applied over the left PFC, more specifically with (b) an anode over the dorsolateral prefrontal cortex (DLPFC) (c) during encoding. We report two new double-blind experiments evaluating the efficacy of this type of brain stimulation on verbal episodic memory, whilst improving on some methodological aspects of previous research.

Several studies have looked at the effect of left PFC-tDCS effects on verbal recognition memory. In their single-blind experiment, Javadi and Walsh [3] investigated the memory performance of healthy young adults following intentional learning of visually presented words. Compared to baseline, anodal stimulation significantly increased, and cathodal stimulation significantly decreased hit rates in an old–new recognition task an hour after stimulation, leading the authors to suggest the potential of tDCS to modulate encoding-related left prefrontal cortical activity and corresponding behaviour in a polarity-specific fashion. Corroborating these findings, similar results were reported from a related experiment [4] evaluating the effect of short-duration (1.6 s) stimulation.

However, further results on improved verbal recognition have not been forthcoming since. Studying the effects of spaced repetition on learning efficiency, Manuel and Schnider [11] adopted similar learning and stimulation parameters investigating the effects of stimulation over PFC and PPC during learning and measured verbal memory in a subsequent recognition task. Testing retrieval in healthy young participants following learning with either active-anodal-left, sham-anodal-left, or active-anodal-right stimulation, they reported no modulation of spacing effects and no enhancement following anodal PFC-tDCS versus sham. Given potentially contradicting findings on the topic, independent replication may be beneficial.

The three experiments described above [3,4,11] provided a memory measure based on the proportion of correctly identified items. Whilst this measure may reflect changes in underlying memory, dual response decision tasks such as the old–new memory paradigm could be influenced by response bias (Signal Detection Theory; [12]). Sensitivity measures based on this theory would take into account changes to both true and false positives, providing a measure of improvement of detection: Identifying true memories and correcting for response bias. In addition, same-day effects on memory could be investigated on a longer timescale to evaluate potential real-life utility. Finally, positive findings in the field may benefit from corroborating evidence from double-blind designs.

Some further findings on the potentially beneficial effect of PFC-tDCS stem from experiments in which participants perform the learning task under intentional learning instructions and are tested on recall, as opposed to recognition tasks at retrieval. It has been recently suggested that these experimental methods may be contributing to the success of memory enhancement with PFC-tDCS during encoding [7,13]. Studies adopting these methods have also produced mixed results so far, as seen below.

Sandrini et al. [6] evaluated the effects of left-PFC tDCS during encoding on subsequent memory specifically in older adults. Active as opposed to sham-group participants performed better in a free recall task of previously learned word lists 48 h after stimulation. It has been suggested that such effects outlasting the stimulation period could be particularly promising for neurorehabilitation purposes [14]. However, some studies have reported no enhancement effects on verbal recall. Nikolin et al. [15] applied tDCS during a verbal learning task and compared the effects of high-definition electrode placements aiming to target either the left dorsolateral prefrontal cortex, planum temporale, or medial temporal lobe, with a sham as the control. The online effects of DLPFC stimulation versus the sham appeared to enhance the speed of learning, but there were no differences in 20 min delayed recall. In a further study, Habich et al. [16] investigated the effect of left PFC versus sham stimulation on learning visually presented nouns. Healthy young participants performed immediate recall tasks whilst undergoing stimulation, followed by a 15 min delayed retrieval test. The authors reported potential positive effects for people with lower initial performance, but no enhancement effects in the delayed recall task 15–30 min after encoding. In addition, Brunye et al. [17] reported significantly worse performance in 2-day delayed verbal recall following encoding in the anodal condition, relative to both cathodal and sham stimulation over PFC.

Thus, it appears that whilst there are some positive results on the effect of left-PFC tDCS during intentional learning on subsequent verbal recall, contradictory and inconclusive findings also exist, similarly to the case of verbal recognition discussed earlier. Correspondingly, we note that the use of cumulative evidence (e.g., meta-analyses) is increasing but still relatively new in the field of neuromodulation. Conclusions from these works are somewhat split as to whether they suggest tDCS-enhancement effects in various cognitive domains (e.g., [18,19], or specifically for elderly participants: [20]) or the contrary, suggesting a lack of evidence for effectiveness (e.g., [21,22,23]). Additional experiments and replications of earlier works would be key in providing an accurate understanding of the generalisability and limitation of any effects previously reported.

Both local and global neural effects have been previously proposed as plausible mechanisms of neural and behavioural enhancement. Initial work on the technique [24] demonstrated that neural activity in the motor cortex can undergo subtle modulation depending on the polarity of the electrode over the area: Excitatory in the case of anodal and inhibitory in the case of cathodal stimulation. Correspondingly, some studies exploring the potential positive effects of tDCS on verbal memory described earlier [3,6,25,26] hypothesised that modulation is, at least in part, due to this polarity-based shift in neuronal excitability, i.e., facilitatory effects in the proximity of the anodal electrode over the left DLPFC. At the same time, due to the presence of a cathodal electrode over a contralateral frontal (supraorbital) area, stimulation may be best described as bipolar, where the placement of both electrodes affects current distribution and, thus, the actual target area. In addition to this modulation in the relative proximity of electrode sites, tDCS is also described as having more global potential effects, e.g., on the activity of distant but functionally connected areas [27,28]. Specifically, in the case of the anode over DLPFC (F3) and the cathode over the right supraorbital montage as in this study, changes in the Default Mode Network (DMN) and Fronto-Parietal Network (FPN) have previously been reported [29]. In addition, a study combining PFC-tDCS with large-scale neurophysiological recordings from monkeys showed an improvement in memory performance [30]. While firing rates did not change within the stimulated area, tDCS induced large, low-frequency oscillations in the underlying tissue. These findings are consistent with the suggestion that tDCS induces functional changes by dynamic modulation of functional connectivity [27,28,29].

Our current aim is to address previous conflicting results regarding the efficacy of left-PFC tDCS on subsequent verbal episodic memory and recognition. The choice of stimulation parameters has been guided by studies reporting potentially successful memory enhancement via pre-frontal stimulation. Thus, we aimed to closely replicate the montage from previous studies [3,6,25,26] using an F3-contralateral supraorbital electrode pair, aiming to target the left PFC with 1.5 mA stimulation intensity. Nevertheless, we note that different levels of current strength may play an important role in the efficacy of neural modulation. For example, higher intensities have been shown to be more successful in enhancing motor-evoked potentials [31] or both performance and regional blood flow in a finger-tapping task [32], even if the dose response has not always been found to be linear [33,34]. tDCS delivered offline at various key stages (e.g., during consolidation, reconsolidation, or offline stimulation prior to encoding) may plausibly affect memory. We chose an online (concurrent) stimulation specifically in order to allow testing our main hypothesis: Whether neuromodulation during encoding would have a replicable, positive effect on subsequent memory.

In Experiment 1, we investigated whether left PFC-tDCS applied during incidental learning has a positive effect on verbal recognition memory, measured in same-day and next-day retrieval. Our aim was to reduce the potential bias by adopting a more reliable recognition memory measure (*d*′) rarely used in similar experiments so far, testing potential enhancement effects in the short- and longer-term and adopting a double-blind design in order to reduce the possibility of bias.

Recent theoretical suggestions regarding the effect of stimulation on verbal memory linked to intentional learning at encoding [7] or adopting a recall test during retrieval [13] were addressed in Experiment 2. In this separate double-blind experiment, potential positive effects of stimulation on verbal long-term memory were evaluated whilst applied during intentional learning instruction and tested with same-day and next-day delayed recall and recognition tasks. Associative memory tasks not related to these aims were also tested.

The key difference between Experiments 1 and 2 was in learning instructions (incidental versus intentional) and primary outcomes (recognition versus free recall). Both experiments were 2 × 2 mixed designs where the main effect of the stimulation (between-subjects: Active versus sham), the main effect of time (within-subjects: Same day, next day), and their interaction were modelled. We predicted that active tDCS would enhance memory performance relative to the sham, and memory performance would decline over time. In addition, we also report a cumulative effect size estimate based on previous research and the two experimental reports below.

## 2. Experiment 1: Materials and Methods

### 2.1. Participants

#### 2.1.1. Sample Size

The selection of the sample size was informed by [21] due to the similarity of study design: A large day-dependent modulation of the effect of stimulation: F(2,52) = 3.37, *η*^2^ = 0.12. Using a lower, medium-sized effect expectation (*η*^2^ = 0.06) in G-Power 3.1 [35], N = 40 participants were necessary for a power of *α* = 0.05, *β* = 0.90 for mixed ANOVA. Thus, 40 right-handed, native-English speaking, 18–35-year-old participants were recruited and randomly allocated to groups using a list of stimulation codes generated for the use of the blind experimenter.

#### 2.1.2. Demographic Characteristics

The Active group (N = 19) had 15 female and 4 male participants with a mean age of 19.95 years (SD = 1.65) and a right-handedness score of 83.84 (SD = 22.58). In the sham group (N = 21), there were 17 females and 4 males, the mean age was 19.65 years (SD = 1.63), and the mean right-handedness score was 88.79 (SD = 19.67). Two participants (Active group) did not return to complete the Day 2 memory test.

### 2.2. Materials

#### 2.2.1. Questionnaires

A modified version of the TMS Adult Safety Screen [36] was used as the Screening Questionnaire, ensuring that people with counter-indications to neurostimulation (e.g., epilepsy, brain injury, current psychiatric medication) were identified and excluded. Handedness was measured using the Edinburgh Handedness Inventory (EHI; [37]). Participants reported the perceived degree and timing of sensations of discomfort and side-effects using the ‘Survey of sensations related to transcranial electrical stimulation’ based on [38]. The calculation of scores on questionnaires is described in detail in Appendix A.

#### 2.2.2. Experimental Stimuli

Four lists of concrete words (N = 180 each) were selected from the MRC Psycholinguistic Database [39,40], each containing an equal number of items describing either living (e.g., teacher, rabbit) or non-living entities (table, railway). Lists were created to comprise items similar in terms of concreteness, imageability, and length, as described in Appendix A. The lengths of the lists were set based on piloting to avoid ceiling effects in the subsequent memory task. Lists were used in a counterbalanced fashion, thus all words had an equal probability to be presented (a) as a target or non-target and to be used in (b) Day 1 or Day 2 retrieval testing.

#### 2.2.3. Brain Stimulator

A Neuroconn DC stimulator (NeuroCare Group Gmbh, Muenchen, Germany) was used to administer current to the brain. In the active tDCS condition, the tDCS stimulator was set to administer 1.5 mA up for 16 min (covering a de-sensitisation period and the duration of the learning task) with both ramp-up and ramp-down times set to 20 s. Electrode sizes of 5 × 7 cm^2^ ensured that the density of the delivered current (0.043 mA/cm^2^) remained below commonly accepted safety limits [41]. In the sham tDCS condition, the stimulator delivered a 1.5 mA direct current for 30 s, with a similar 20 s ramp-up and ramp-down period as the real stimulation. Furthermore, 5 × 7 cm^2^ electrode sponges soaked in saline solution were prepared individually for each testing session. Aqueous Potassium Chloride (KCL, Scientific and Chemical Supplies Ltd., Bilston, UK, Cat. No: PO035) was used with an approximate concentration of 100 mM.

### 2.3. Procedure

#### 2.3.1. Electrode Setup

Participants were seated in an experimental cubicle for both days’ testing sessions. Participants’ scalps were inspected for any lesions prior to setup. If no problems were found, the experimenter proceeded to identify target sites for the electrodes according to the international 10–20 electroencephalogram international system for electrode placement [42]. The cathode was placed over the right supraorbital area defined as FP2 and the anode over DLPFC defined as F3 [43]. The sponges were placed vertically, i.e., their longer midline axes were aligned approximately towards Cz. The electrodes were secured in their position in the proximity of the scalp and held in place using a rubber hairband. The impedance of the stimulation setup was checked before stimulation and kept under 10 kΩ. The stimulator was programmed to deliver 16 min of stimulation. This allowed a several-minute-long desensitisation period before the commencement of the experimental task.

#### 2.3.2. Incidental Learning Task

The experimenter described that all three tasks are word categorization tasks, without revealing the use of memory tests later. In all tasks, participants were instructed to respond as quickly and accurately as possible. All word stimuli were presented on a computer screen using E-Prime 2.0 software, presented on a 22″ HP LCD screen. Participants were presented with a list of 180 words and asked to indicate for each item whether they referred to something living or not living. This was performed by pressing a key using their right hand. Each word was presented for 1 s, with a variable 2–4 s inter-trial interval, taking 12 min in total. An example trial is shown in Figure 1 below. The task was followed by a 20 min resting period, which also included the completion of the Sensations Questionnaire.

#### 2.3.3. Recognition Task, Day 1

Participants were presented with a list of words (N = 180), with half of the items drawn from those presented in the incidental learning task (‘old words’), and the other half were previously unseen items (‘new words’). Words remained on the screen for 2 s, with a randomly jittered inter-trial interval of 2–4 s, taking a total of 15 min. Participants’ task was to indicate whether the word was ‘old’ (presented during the learning task) or ‘new’ (not previously presented). This task assessed short-term recognition performance. Following the test, participants left and were asked to return the following day to perform a further word categorisation task.

#### 2.3.4. Recognition Task, Day 2

The procedure of the recognition task from day one was repeated using previously untested target and non-target words (N = 90 each). This task assessed long-term retention and recognition performance. On completion of testing, participants were asked to make a guess regarding their randomisation status, and the experimenter retrieved the randomisation status from a sealed envelope and debriefed the participant.

### 2.4. Data Analysis

Inferential statistical analysis was carried out using R [44], including the *afex* package [45] for ANOVA and *metafor* [46] for calculating a cumulative effect. Analysis scripts are available at [URL to be confirmed in published article] with openly accessible data at [URL to be confirmed in published article].

## 3. Experiment 1: Results

### 3.1. Memory Performance Measure

Each trial with a valid response was categorised as one of the following four outcomes: Hit (old word correctly identified as such), miss (old word classified as new), false positive (a new word identified as old), or correct rejection (a new word correctly classified). For both recall sessions on Day 1 and Day 2, the hit rate was calculated as a proportion of target words correctly identified as ‘OLD’, yielding a value between 0 and 1. Similarly, for false alarms, the proportion of non-target words incorrectly identified as ‘OLD’ was calculated. D-prime (Discriminability Index: *d*′) was selected a priori as the main outcome variable, in order to measure performance changes factoring in Receiver-Operator characteristics including response bias. *d*′ was calculated using standardised scores of hit rates and false alarms [47], measuring sensitivity to detect true memories amongst all stimuli:*d*′ = *zHR* − *zFA*
for Days 1 and 2, respectively, for each participant. The discrimination performance of all participants was higher than chance level. Due to multiple testing (×3 *a priori* analyses: *d*′, hit rates, false alarms), we corrected for familywise error using a Bonferroni correction resulting in an alpha value of 0.017 (0.05/3) for the interpretation of results.

### 3.2. D-Prime (d′)

A 2 × 2 mixed ANOVA was performed with *d*′ as the a priori dependent variable. The main effect of time was significant, F(1,35) = 179.08, *η*^2^ = 0.837, *p* < 0.001. Mean performance was better on Day 1 (*d*′ = 1.47, 95%CI: 1.29 to 1.64) than on Day 2 (*d*′ = 0.81, 95%CI: 0.68 to 0.93). The main effect of Stimulation Group was not significant, F(1,35) = 0.15, *η*^2^ = 0.004, *p* = 0.703. The mean performances in the Active and Sham groups were *d*′ = 1.16 (95%CI: 0.95 to 1.38) and *d*′ = 1.11 (95%CI: 0.91 to 1.30), respectively. The interaction of Time and Stimulation Group, as shown in Figure 2, was not significant, F(1,35) = 1.10, *η*^2^ = 0.03, *p* = 0.302.

### 3.3. Hit Rates (HR)

The main effect of time on hit rates was significant, F(1,35) = 101.71, *η*^2^ = 0.744, *p* < 0.001. A higher proportion of items was correctly identified as ‘Old’ on Day 1 (HR = 0.74, (95%CI: 0.70 to 0.078) than on Day 2 (HR = 0.56, 95%CI: 0.50 to 0.61). The main effect of the Stimulation Group was not significant, F(1,35) = 0.16, *η*^2^ = 0.004, *p* = 0.695. The mean performances in the Active and Sham groups were HR = 0.64 (95%CI: 0.58 to 0.71) and HR = 0.66 (95%CI: 0.60 to 0.72), respectively. The interaction between Time and Stimulation Group was not significant, F(1,35) = 3.59, *η*^2^ = 0.093, *p* = 0.067, as plotted in Figure 3.

### 3.4. False Alarms (FA)

The main effect of time on false alarms was not significant, F(1,35) = 2.91, *η*^2^ = 0.08, *p* = 0.097. False alarm rates slightly increased from Day 1 (FA = 0.24, 95%CI: 0.20 to 0.28) to Day 2 (FA = 0.27, 95%CI: 0.23 to 0.31). The main effect of Stimulation Group was not significant, F(1,35) = 0.77, *η*^2^ = 0.021, *p* = 0.388. False alarm rates in the Active and Sham groups were FA = 0.24 (95%CI: 0.19 to 0.29) and FA = 0.27 (95%CI: 0.23 to 0.32), respectively. The interaction between Time and Stimulation Group was not significant, F(1,35) = 1.53, *η*^2^ = 0.042, *p* = 0.224, as seen in Figure 4.

### 3.5. Post-Hoc Exploratory Analysis of Signal Detection Threshold (c)

Potential modulation of signal detection threshold by group status or testing day was explored by calculating c:*c* = −[*z*(*H*) + *z*(*F*)]/2.

Higher (positive) values indicate more conservative and lower (negative_) values more liberal memory decision criteria. The main effect of time on the detection threshold was significant, F(1,35) = 22.73, *η*^2^ = 0.394, *p* < 0.001. The threshold increased from Day 1 (*c* = 0.02, 95%CI: −0.08 to 0.12) to Day 2 (*c* = 0.25, 95%CI: 0.13 to 0.37). The main effect of Stimulation Group was not significant, F(1,35) = 0.091, *η*^2^ = 0.025, *p* = 0.347. Mean values in the Active and Sham groups were *c* = 0.18 (95%CI: 0.04 to 0.32) and *c* = 0.09 (95%CI: −0.04 to 0.22), respectively. The interaction between Time and Stimulation Group was significant, F(1,35) = 4.26, *η*^2^ = 0.109, *p* = 0.046, as seen in Figure 5. Thus, there appeared to be a change in the detection threshold from Day 1 to Day 2, potentially influenced by group status on Day 1. A follow-up *t*-test of the Day 1 threshold was indicative of potential differences, i.e., a lower (more liberal) threshold in the Active versus Sham group: *t*(37) = −2.415, *p* = 0.021, Cohen’s *d* = −0.77 (95%CI: −1.44 to −0.10).

### 3.6. Survey of Sensations Related to Transcranial Electrical Stimulation Results

Total sensation scores reported by participants after the experiment were M = 4.14 with a 95%CI [2.28 to 6.01] in the sham and M = 4.94 with a 95%CI [3.45 to 6.43] in the active condition. The mean difference between groups was M(*dif*) = 0.8 with a 95%CI [−3.05 to 1.46], with zero difference being within the estimated range.

Guesses regarding allocation status were available for 29 participants and are reported in Table 1 below. In order to evaluate blinding success, the Bang Blinding Index (BBI) [48,49] was calculated based on active/sham/don’t know response options. The BBI allows for evaluating the proportion of participants guessing their allocation status correctly above chance, with values 1 and −1 indicating total unblinding or total opposite guessing, respectively, and 0 value corresponding with chance-level guessing. BBI was calculated in terms of the proportion of guesses of allocation to the active condition with a Bonferroni correction to account for evaluating two experimental groups. Based on the estimate intervals in Table 1, both groups appeared to be guessing their allocation status to be the active stimulation condition above chance levels (positive values), with no significant difference between groups in this behaviour (overlap). This indicates a potential similarity in the sham condition experience to active stimulation.

## 4. Experiment 2: Materials and Methods

### 4.1. Participants

#### 4.1.1. Sample Size Calculation

Sample size calculation using G-Power 3.1 [35] was based on the expectation of a medium-sized population effect (Cohen’s *f* = 0.25, *r* = 0.55) and a 2-by-2 mixed ANOVA design. Using values of *α* = 0.05, and *β* = 0.90 in order to control for Type I and Type II errors, N = 40 participants were set as a recruitment target split into two equal groups.

#### 4.1.2. Demographic Statistics

All participants were right-handed, native English speakers between 18 and 35 years of age, recruited independently from the sample in Experiment 1. Participants were pre-screened for suitability and safety before stimulation (see below). As in the previous experiment, all participants were screened for right-handedness. Out of a total of N = 49 participants, 3 were excluded after screening (left-handedness: 1, tDCS-safety screening fail: 1, and withdrawing before the task: 1), and 3 due to missing responses or chance-level performance. One further dataset could not be analysed due to a data saving error, leaving N = 42 for the analysis reported below. The Active group consisted of 20 participants (18 female, 2 male) with a mean age of 19.85 years (SD = 1.73). In the Sham group, there were 22 participants (17 female, 5 male) with a mean age of 19.33 years (SD = 1.39).

#### 4.1.3. Randomisation Procedure

The experimental testing was performed in a double-blind fashion. A random sequence was generated using Matlab software, ensuring an equal probability of allocation to the active or sham condition in each subsequent block of 10 participants. The experimenter carrying out testing was blind to the allocation status of the participant until the debrief, for which the allocation status was obtained from a sealed envelope. The allocation status of participants was not concealed during data analysis.

### 4.2. Materials

#### 4.2.1. Questionnaires

The screening questionnaire (TASS), handedness inventory (EHI), and sensations questionnaire were used as described in Experiment 1.

#### 4.2.2. Word Lists

Words of comparable length (2–3 syllables, 6–10 letters) and high levels of imageability, concreteness, and frequency were selected from the MRC Psycholinguistic Database [39,40]. Selected items were allocated into two groups with similar characteristics along these attributes, as described in Appendix A, and presented in a counterbalanced order so that items had an equal chance of being presented as a target or non-target in recognition tasks. The length of word lists was determined following the piloting of the experimental tasks. Pilot participants typically remembered 15–20 items of the 40 targets learned on Day 1, i.e., this length of the stimuli set allowed for avoiding floor- or ceiling effects. All word stimuli were presented on a computer screen using E-Prime 2.0 software, presented on a 22″ HP LCD screen.

#### 4.2.3. Stimulation Equipment and Parameters

The Neuroconn DC stimulator (NeuroCare Group Gmbh, Muenchen, Germany) was programmed to deliver active or sham stimulation identical to the method described above in Experiment 1.

### 4.3. Procedure

#### 4.3.1. Consent and Screening

At the beginning of the session, participants gave informed written consent and had the opportunity to ask any questions. Participants completed the screening questionnaire, and their eligibility was checked before any further preparation took place. Following consent and screening, participants were seated in a designated testing cubicle where the rest of the testing took place.

#### 4.3.2. Electrode Setup

The use of the conductive solution, head measurements, electrode positioning, and adjustments was performed similarly to the method described in Experiment 1, including the use of an acclimatisation period prior to the commencement of the experimental tasks.

#### 4.3.3. Procedure Overview

An indicative timeline of the experimental tasks is displayed below in Figure 6. Participants were seated approximately 100 cm from the screen. Participants were explicitly told to memorise words for later retrieval. In all tasks, participants were instructed to respond as quickly and accurately as possible.

#### 4.3.4. Intentional Learning Task

In each block, a list of 40 words was displayed in a random order using E-Prime 2.0 on an HP 14″ LCD monitor. This was repeated twice (3 blocks in total) with a short, self-timed break between blocks. Words were presented in a font size of 18, type Courier New, on a black background in one of the following colours: Red, blue, yellow, or green. The trial duration was 2 s, with an inter-trial interval with a mean duration of 3 s, and a random duration in the range of 2.7–3.3 s. Four colour lists were created so words had an equal chance (across participants) of appearing in any of the four colours. Participants were instructed to memorise the words and corresponding presentation colours and say the word and corresponding colour out loud during each trial. This task was adopted to ensure participants were aware of a subsequent memory task (intentional learning) and to allow an additional, associative memory test to be performed.

#### 4.3.5. Resting Period

This period lasted 20 min following the learning task. During this time, the experimenter removed the tDCS equipment. Participants completed the tDCS sensations questionnaire and had an opportunity to take a short break.

#### 4.3.6. Recall Task, Day 1 and Day 2

Following the rest period, participants were instructed to recall as many words as they could remember from the learning task. A pen and paper were provided, and a maximum of 5 min was allowed for the completion of the task. The answer sheet was checked by the experimenter, and the number of words recalled was recorded. The accuracy of the items was checked following the session. The test was repeated in the same format in the Day 2 testing session.

#### 4.3.7. Associative Recall Task, Day 1 and Day 2

The experimenter handed back the word recall sheet to the participant and asked them to identify the colours associated with each word by writing them on the sheet next to the recalled word item. The same procedure was repeated on Day 2.

#### 4.3.8. Word Recognition Task, Day 2

Eighty words (40 targets and 40 fillers) were presented for 2 s each using E-Prime 2.0. The inter-trial interval ranged from 3.7 to 4.3 s, with a mean duration of 4 s. Participants were instructed to indicate with a key press whether they remembered the word from day one (old word) or not (new word).

#### 4.3.9. Colour Recognition Task, Day 2

Forty target words were presented in white font on a black background for 2 s each using E-Prime 2.0. All four colour options (red, blue, green, and yellow) were presented at the bottom of the display. Participants performed a forced-choice colour recognition task to indicate the colour associated with words in the encoding phase. Answer options remained on the screen for a further 5 s. The inter-trial interval was 2 s (a range of 1.7–2.3 s).

#### 4.3.10. Debrief

Following the recognition tasks on Day 2, the experimenter opened the randomisation envelope, identified the allocation status of the participant, and debriefed them accordingly.

## 5. Experiment 2: Results

### 5.1. Memory Performance Measure

Memory performance was tested twice, i.e., 20 min (Day 1) and one day after stimulation (Day 2). The number of correctly recalled words was selected as the primary dependent variable. In addition, word recognition on Day 2 was measured using *d*′ and calculated as in the previous experiment: *d*′ = *zHR* − *zFA* [47]. Similar to Experiment 1, we corrected for familywise error using a Bonferroni correction resulting in an alpha-threshold of 0.017 (0.05/3) due to the three planned analyses reported below.

### 5.2. Verbal Free Recall

A 2-by-2 mixed ANOVA revealed there was a significant main effect of time: Forgetting occurred as expected in the one-day interval, F(1,40) = 73.81, *η*^2^ = 0.649, *p* < 0.001. The number of recalled words was M = 12.73 (95% CI: 11.38 to 14.09) on day one and M = 10.62 (95% CI: 9.31 to 11.93) on day two. Regarding the main effect of stimulation, participants in the active conditions performed worse (M = 10.35, 95%CI: 8.45 to 12.25) than their sham-group counterparts (M = 13, 95%CI: 11.19 to 14.81), although this effect was not significant at the adjusted alpha level described above, F(1,40) = 4.17, *η*^2^ = 0.094, *p* = 0.048. The performance decrease due to tDCS appeared to be more marked on day one, although this interaction (Figure 7) was also not significant at the adjusted alpha level, F(1,40) = 4.42, *η*^2^ = 0.099, *p* = 0.042.

### 5.3. Word Recognition

Participants in the active tDCS condition had a lower mean *d*′ score (M = 1.65, SD = 0.68) than those in the sham condition (M = 1.77, SD = 0.64) on day two of testing, but there was no indication of a significant difference between the groups based on a *t*-test: *t*(40) = −0.548, *p* = 0.59, Cohen’s *d* = −0.17 (95%CI: −0.79 to 0.46).

### 5.4. Associative Colour Recognition

A similar pattern was found regarding day-two associative colour recognition, with no significant difference between groups, *t*(40) = 0.681, *p* = 0.500, Cohen’s *d* = 0.01 (95%CI: −0.42 to 0.45), with accuracy scores M = 0.48, SD = 0.16 for the active group and M = 0.46, SD = 0.08 for the sham stimulation group.

### 5.5. Post-Hoc Exploratory Analysis of Signal Detection Threshold (c)

Recognition tests were only carried out on Day 2. The *t*-test of threshold criterion *c* did not indicate significant differences between Active (*c* = −0.03, SD = 0.45) and Sham (*c* = 0.03, SD = 0.36) groups, *t*(40) = −0.488, *p* = 0.628, Cohen’s *d* = −0.15 (95%CI: −0.78 to 0.47).

### 5.6. Cumulative Effect Size

Based on the effect of active stimulation with the F3 anode reported in [21], and the main effect of stimulation from Experiments 1 (*d*′) and 2 (recall), we carried out a random-effects meta-analysis using the *metafor* [46] package. This provided a summary effect size estimate (with 95%CI) as *g* = −0.06 (−0.53; 0.40). This indicates moderate negative to small positive population effects of active tDCS setups with the F3 anode versus sham.

## 6. Discussion

In two double-blind randomized, sham-controlled experiments, evidence of an overall positive effect of stimulation was not found. In Experiment 1, participants who completed the incidental encoding task in the active stimulation group were not more successful in discriminating between previously seen versus new verbal items compared to participants in the sham group. These effects did not differ significantly in magnitude between the two days. Whilst blinding appeared to be successful, with participants guessing experimental status similarly in both groups, exploratory analysis suggests a potentially more liberal memory detection threshold in the Active group on the shorter delay (Day 1). In Experiment 2, active prefrontal tDCS during intentional encoding did not appear to improve subsequent free recall performance when compared to sham. In the second study, the tDCS group’s performance was lower than in the control group during free word recall, and this decrease in performance compared to the sham control group appeared more marked on Day 1. No difference was found between groups in second-day tests of word or associative colour recognition. Thus, overall, the experiment did not find evidence for verbal memory enhancement of left prefrontal tDCS applied during encoding. The calculated cumulative effect size of similar active tDCS montages on verbal episodic memory is close to zero, with a confidence interval ranging from moderate negative to small positive values.

In relation to the initial hypothesis that stimulation facilitates episodic memory performance, participants in the active group did not outperform participants in the sham group either in the short- or long-term memory test. Thus, the experiment failed to find a positive effect on verbal recognition memory in this group of healthy young adults. Our findings, stemming from an improved experimental protocol, led to lower enhancement estimates than some previous works [3,4] described earlier. Regarding free recall, the lack of enhancement detected is not inconsistent with other findings in the field. Two of the earlier mentioned studies [15,16] found no significant effect of similar tDCS procedures on long-term verbal recall. Other studies presented results in the opposite direction, e.g., an improvement in recall was found by Sandrini et al. [6] albeit in a different sample (healthy older adults), and a decrease in performance amongst healthy young participants in [17].

Results from the current experiment and the cumulative effect size based on current experiments and previous research are consistent with a population effect ranging from small negative to small positive—and potentially close to null. This estimate is also close to recent meta-analysis results from [50], in which the authors evaluated the effect of anodal and cathodal tDCS on different memory performance measures. In this review, the overall effect of tDCS on memory was found to be small, close to zero in the case of anodal stimulation, with task type and stimulation duration proposed as potential moderators in exploratory analyses. Thus, whilst the notion of applying this form of brain stimulation during learning as a form of memory enhancement is worthy of attention, experimental evidence from multiple studies using a single session of left prefrontal tDCS does not strongly support this possibility. As discussed earlier, potential effects of tDCS have been linked to activity modulation of both specific nodes or more global changes of memory-related neural networks. Whilst our study has not used an active control condition to vary the target of stimulation as a comparison, nor explored neural mechanisms in detail, e.g., with concurrent neuroimaging, this is the subject of a current investigation in our group.

Whilst there is some consensus in the scientific community regarding the potential neural mechanisms of tDCS [2] and that its short-term effects can be reliably demonstrated under laboratory conditions, the degree of transcranial penetration of direct current under safe administration parameters and limits (e.g., as reviewed in [41]) has been recently been questioned. For example, based on intracranial recordings of current density in in vivo animal specimens and human cadavers, some argue [51] that the commonly used current tDCS protocols may not achieve the level of penetration and neural stimulation that present-day models predict. Thus, longer-term effects, as measured in the current experiments, may be more difficult to achieve and detect. Studies validating the current distribution models in vivo (e.g., [52]) could improve our understanding of how specific brain areas could be efficiently targeted in memory enhancement studies.

Our study intended to address several methodological aspects of previous research. A priori outcome measures and a double-blind design were used to reduce potential bias, and previous research was used to reduce the uncertainty of effect-size estimates. At the same time, the use of an active control condition (e.g., stimulation of an alternative target site) would have allowed for confirming whether any effects are specific to stimulation of the target area (e.g., PFC) or potentially due to more generic physiological effects due to the experience of stimulation.

We note that some aspects of our own study design also have potentially stronger alternatives. For example, (1) higher stimulation intensity could potentially lead to a greater degree of neural modulation; (2) adoption of the increasingly popular high-definition montage [53] may allow for more accurate targeting of specific areas of interest; and (3) the use of a cross-over design or a more in-depth account of participant traits and abilities (e.g., cognitive ability and education) would have allowed us to take individual characteristics into account.

A plausible alternative explanation for the lack of memory enhancement also exists: The difference in populations sampled compared to some previous papers. Whilst positive effects of PFC-tDCS during encoding have been reported in healthy young [3] and older [6] adults, or both [7], it has been suggested [16] that amongst young, healthy adults, the positive effects of left PFC stimulation may be specific to participants who initially perform less than others in cognitive tasks. Whilst these initial findings have not yet been replicated successfully, as discussed above, some recent meta-analyses also suggest there may be a small positive effect of tDCS in older adults. For example, Galli et al. [50] estimated a marginally significant, small effect in a sub-group analysis of elderly participants (*g* = 0.41 [0.00; 0.83]) and Huo et al. [20] reported a longer-term effect as *g* = 0.404 [0.22; 0.96] in the same population. Similarly, it has been suggested [50] that a longer stimulation duration and recall tasks may yield a greater performance increase. In addition, closely related stimulation techniques have also been explored recently and demonstrated potential efficacy in this population, for example, transcranial alternating current stimulation (tACS) as compared to tDCS [54], or when applied using a repetitive administration protocol [55]. We note that these findings may be investigated in more depth, albeit they may necessitate sample sizes far larger than what is currently common practice in the field [56].

The technique has a number of parameters (e.g., electrode placement, polarity, intensity, duration, and timing) that can be modified, giving rise to potentially high variability of methodologies between studies and making direct comparisons of evidence more difficult. Identifying replicable effects of tDCS on memory would also allow a systematic investigation of the influence of stimulation parameters. For example, varying the polarity of electrodes or comparing the behavioural and neural effects of different cathodal placement conditions may provide insight into any modulation due to targeting specific areas or anodal/cathodal or bipolar stimulation.

Some authors argued that questionable research practices and the lack of reproducibility or consistency in findings may call into question the efficacy of previous research on tDCS [57] or cognitive neuroscience in general [58]. Therefore, we commend that some authors of studies with novel findings on the cognitive effect of tDCS were also involved in close replication efforts, for example, Medvedeva et al. [7] followed up by Petrovskaya et al. [59] or Habich et al. [16] by Habich et al. [60]. We note that, similarly to our study, these two follow-up studies did not replicate the large positive effects found in the original papers.

It may be worth noting that certain methodological and statistical aspects of previous literature could indeed be improved. Currently, there is much interest in tDCS research, and a large number of tDCS studies investigating a variety of outcome measures are being published yearly. At the same time, pre-publication of analysis protocols, accounting for multiple statistical comparisons, and employing larger sample sizes is not universal; a priori analytical intentions and the extent of experimenter blinding and randomisation protocols are not always fully accessible. The increased use of double-blind designs, randomized, controlled trials, open-science initiatives such as pre-registration [61,62], and data sharing may improve the transparency, accuracy, and reliability of experimental reporting in the field [63,64]. This, in turn, will allow a more accurate investigation of the real-world potential of tDCS to improve human cognitive performance.

## Figures and Tables

**Figure 1 brainsci-13-00241-f001:**
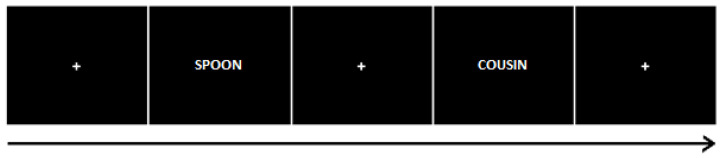
Incidental encoding and recognition task structure: Short presentation of words interleaved with a random duration (2.5 to 3.5 s) fixation cross. Word presentation times were 1 s (encoding task) or 2 s (recognition task).

**Figure 2 brainsci-13-00241-f002:**
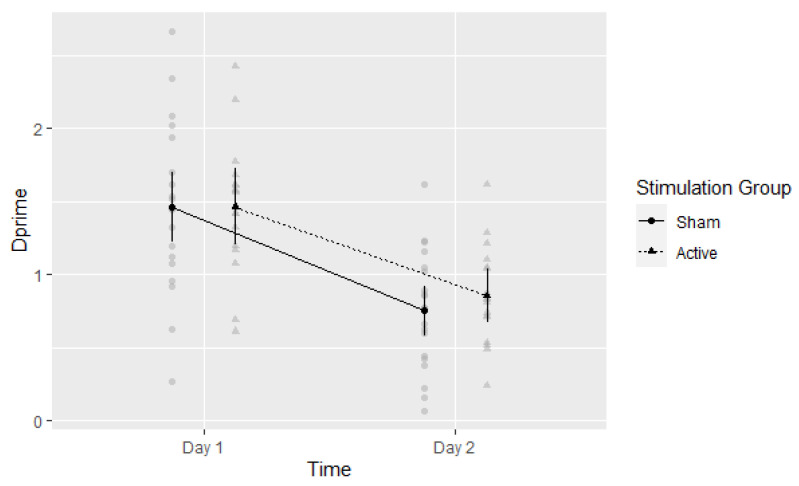
Experiment 1: Discrimination performance per group per time point. Means with 95% CI plotted. Gray circles and triangles indicate individual values: Sham and Active respectively.

**Figure 3 brainsci-13-00241-f003:**
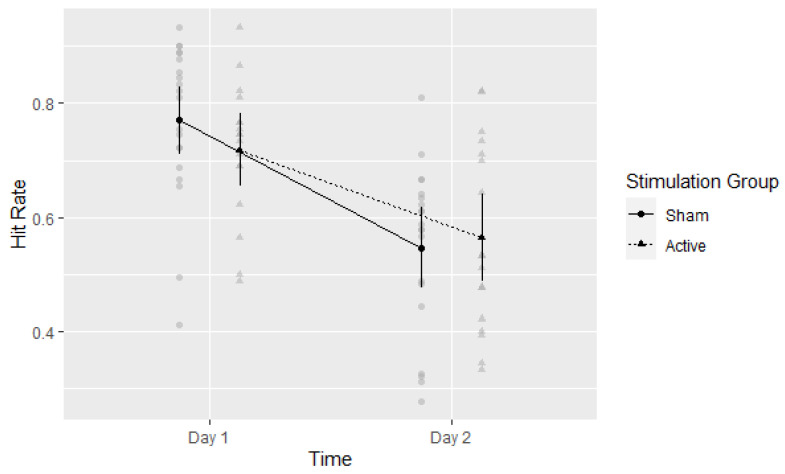
Experiment 1: Hit rate per group per time point. Means with 95% CI plotted. Gray circles and triangles indicate individual values: Sham and Active respectively.

**Figure 4 brainsci-13-00241-f004:**
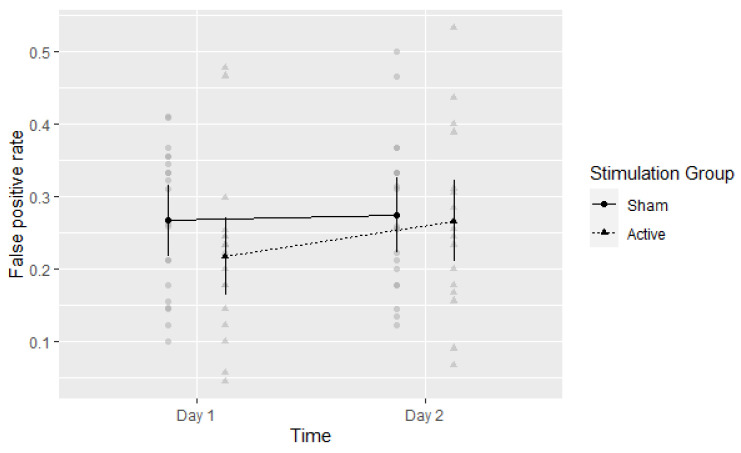
Experiment 1: False positive rate per group per time point. Means with 95% CI plotted. Gray circles and triangles indicate individual values: Sham and Active respectively.

**Figure 5 brainsci-13-00241-f005:**
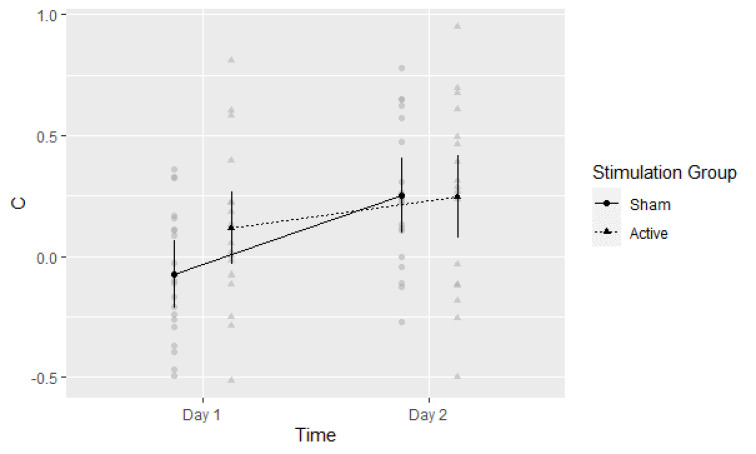
Experiment 1: Response bias threshold *c* per group per time point. Means with 95% CI plotted. Gray circles and triangles indicate individual values: Sham and Active respectively.

**Figure 6 brainsci-13-00241-f006:**
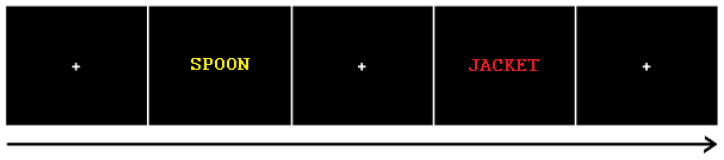
Intentional encoding and recognition task structure: Short presentation of words interleaved with a random duration (2.5 to 3.5 s) fixation cross. Word presentation times were 1 s (encoding task) or 2 s (recognition task).

**Figure 7 brainsci-13-00241-f007:**
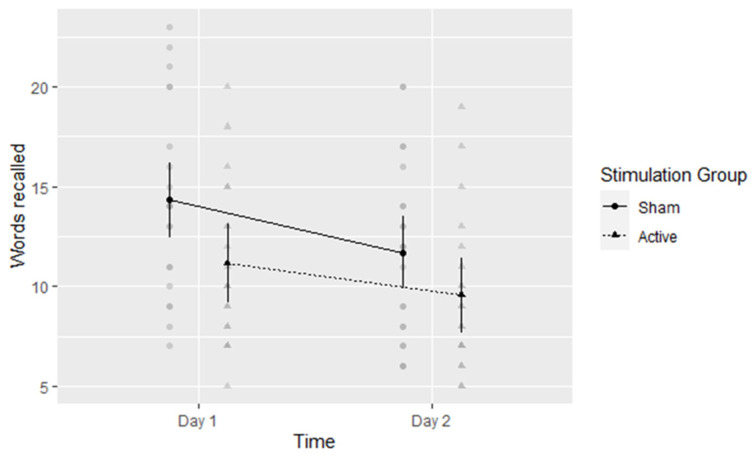
Experiment 2: Number of words recalled per group per time point. Means with 95% CI plotted. Gray circles and triangles indicate individual values: Sham and Active respectively.

**Table 1 brainsci-13-00241-t001:** Participants’ guesses regarding allocation status and BBI per group.

Allocation Guess	Active Group	Sham Group
‘Active’	10	8
‘Sham’	5	5
‘Don’t know’	0	1
BBI [95% CI]	0.333 [0.192 to 0.474]	0.214 [0.064 to 0.365]

## Data Availability

Data reported in this paper are available via the following URL: [insert both datasets on publication] DOI: 10.13140/RG.2.2.34552.52488 and DOI: 10.13140/RG.2.2.12742.14401, together with the R script of the analysis via DOI: 10.13140/RG.2.2.19283.86568.

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
