# Peer review of "Left Prefrontal tDCS during Learning Does Not Enhance Subsequent Verbal Episodic Memory in Young Adults: Results from Two Double-Blind and Sham-Controlled Experiments"

_brainsci, 2023, doi:10.3390/brainsci13020241_

Round 1

Reviewer 1 Report

Overview: Two experiments examined the effects of tDCS targeting the left-lateralized PFC on verbal episodic memory. In experiment 1, the authors examined incidental learning and a recognition test, and in experiment 2 they examined intentional learning and recognition and recall tests. Across experiments and measures, there was no compelling evidence that active versus sham tDCS modulated any aspect of incidental or intentional memory when tested at a short or extended interval.

Overall Assessment: I enjoyed reading this brief article and thought the paper was generally well-written and the experiments were well designed and conducted. I have no major concerns about the paper, but several moderate suggestions for improvement, detailed below.

1. As it stands, the manuscript is very descriptive and applied, and could benefit from more theoretical background. Specifically, additional discussion of the mechanistic models of tDCS effects on the micro, macro, and meso levels of analysis. How do these models relate to the hypothesized effects of tDCS on memory encoding? What is known and unknown in these models, and how might that influence the results? 

2. Non-invasive transcranial electrical stimulation has myriad parameters that can be altered and likely influence effects on performance. Please be very forthcoming about why particular parameters were selected and used herein. For example, the intensity, duration, and online/offline timing of stimulation need to be justified. Further, also reveal why you chose this particular the electrode type and positioning (i.e., F3 and right supraorbital).

3. In addition to the measures reported (e.g., d-prime, hit rates, false alarm rates), please provide full analyses for response bias (e.g., using "c"). Also, please provide full analyses for response times, assuming these data were collected on the recognition tasks.

4. In the analyses, I suggest selecting a single statistical test type rather than switching between ANOVAs and t-tests. However, if you are going to switch between these analysis types, please provide an effect size measure alongside t-test results (e.g., Cohen's d). A more powerful overall analysis approach could be linear mixed effects modeling to account for random and fixed effects.

5. The authors revealed stimulation condition to participants after asking that they guess which group they were in; please detail whether participants were successful in their guessing, which speaks to potential placebo effects in (other) studies.

6. When discussing the present results and how they differ from previous results, the authors should be careful in their language. For example, stating that their results are "in contrast with some previous results" is not entirely true. To be in contrast they would need to show the opposite pattern, i.e., performance decrements. Instead, they are simply not in support of the earlier findings. 

Minor Comments

When discussing the Bonferroni correction, please clarify that you are modifying the alpha criterion to 0.017, not the p-value.

Please define somewhere in the paper the meaning of d-prime (i.e., discriminability index) so the reader is aware why this measure is superior to other measures.

On page 2, line 90+, when discussing the Brunye et al paper, note that anodal impaired recall relative to both cathodal and sham. Please reword.

In section 2.1, Sample Size, the eta-squared value is incorrect (i.e., it should not equal zero). 

On page 4, lines 193+, please describe how the new words were selected for use in this experiment.

On page 5, lines 201+, please reveal whether the order of the two word lists was balanced between days 1 and 2 (i.e., did half of the participants see List 1 on Day 1, and half saw List 2?).

On page 5, line 227, please fix agreement: "performance of all participants was higher..."

In all results, when parenthetically providing means (e.g., mean hit rate, mean d-prime, mean false alarms) please also provide a measure of variance.

On page 8, please justify the use of this specific intentional learning task.

On page 9, lines 369-370, please fix agreement: "The number of recalled words was..."

In the discussion, page 11, lines 438+, the authors suggest that tDCS protocols may not achieve "direct neural stimulation" - however, I do not believe anyone would suggest direct stimulation with tDCS, given that it is a subthreshold neuromodulation technique. Please reword.

Reviewer 2 Report

This is thoughtful study on a relevant issue, that tempers down the possibility to use tDCS as a memory enhancer.

Among the discussed points, I see a missed points, that concerns the bipolar stimulation. Indeed, the anode was positioned on the DLPFC and the cathode was positioned over the right supraorbital area, in both cases using relatively large electrodes. This does not guarantee, of course, that only anodal stimulation might responsible of the experimental negative results. 

I guess that that the whole discussion, that is based on previous contrasting findings of tDCS on the memory performance, should be revised according to this methodological fundamental question. 

Another point to be considered, again in view of the contrasting findings of tDCS on the memory performance, is the type of material to be remembered (i.e., verbal, visuospatial and so on). Indeed, the depth of the memory encoding may vary according to such variable, and consequently also the effects of tDCS cab vary (i.e., a deeper encoding could be likely more difficult to be modified by imperceptible currents delivered on the scalp)

Reviewer 3 Report

In this commendable study the authors evaluated the effect of anodal tDCS on two memory paradigms. While the study design is not novel it is a very thorough replication effort, which is very much needed in sight of the inconsistent findings in the tDCS-based memory enhancement literature. The study also distinguishes itself through being very well-rounded and well-informed (see e.g. sample size calculations, 2 different memory paradigms with piloting, very well anchored in previous literature) and comprehensively written.

While the study found no significant differences between the anodal and sham tDCS conditions in the two memory paradigms (after correction for multiple comparisons), it is undeniably valuable for the brain stimulation community to realistically assess the presence/absence of tDCS benefits.

The only main drawback of the study is that in its replication effort, it does not follow more recent guidelines of stimulation protocols (cross-over design, active control, more high-definition set-up, higher stimulation intensities, etc.). Hence, I would appreciate that the null-findings are also put in this context in the discussion.

I have only a few additional minor comments, outlined below:

Methods – Experiment 1: How many lists of concrete words were created in total? Were all participants presented with the same lists of words for encoding and recognition?

Results – Experiment 1 & 2: Were participants able to guess their group allocation?

Discussion: Generally, it would have been nice to characterize participants a bit better beforehand, education level, performance in another memory test., etc. This might be worth discussing.

Round 2

Reviewer 2 Report

Authors addressed all the points raised